# Work-related smartphone use during off-job hours and work-life conflict: A scoping review

**Holly Blake** [ID] [1,2]*, **Juliet Hassard**[3], **Jasmeet Singh**[4], **Kevin Teoh** [ID] [5]

**1** School of Health Sciences, University of Nottingham, Nottingham, United Kingdom, **2** NIHR Nottingham Biomedical Research Centre, Nottingham, United Kingdom, **3** Queen's Business School, Queen's University Belfast, Belfast, Northern Ireland, **4** Department of Psychology, Nottingham Trent University, Nottingham, United Kingdom, **5** Birkbeck Business School, Birkbeck University of London, London, United Kingdom

* holly.blake@nottingham.ac.uk

**Data Availability Statement:** All relevant data are within the manuscript and its Supporting Information files.

## Abstract

Over recent decades the use of smartphones for work purposes has burgeoned both within and beyond working hours. The aim of the study was to conduct a scoping review to explore the association between the use of smartphone technology for work purposes in off-job hours with employees' self-reported work-life conflict. Arksey and O'Malley's methodological framework was adopted. Searches were conducted in PsycINFO, International Bibliography of the Social Sciences (IBSS), Academic Search Complete, ProQuest Central, Web of Science, ProQuest Theses, Emerald, Business Source Complete, ScienceDirect, Scopus, Google Scholar. Articles were eligible that reported on a sample of workers, were published in English between 1st January 2012 and 29th November 2023. The review was conducted and reported using a quality assessment checklist and PRISMA-ScR (Preferred Reporting Items for Systematic reviews and Meta-Analyses extension for Scoping Reviews). Data charting and synthesis was undertaken narratively, using the framework approach and thematic analysis. Twenty-three studies were identified, conducted in nine countries. Nineteen studies (83%) showed a significant association between increased use of smartphone for work purposes in off job-hours and increased work-life conflict, with small-to-moderate effect sizes. This relationship was mediated by psychological detachment from work, and communication about family demands with one's supervisor. Moderators either strengthened or attenuated the relationship between use of smartphone for work purposes in off job-hours and increased work-life conflict. Findings suggest that smartphone use during off-job hours is likely to impact negatively on work-life conflict, which has implications for employee well-being. Managers could play a key role in clarifying expectations about after-hours availability, reducing job pressure, advocating psychological detachment from work in off-job hours where it is appropriate, and creating a workplace culture where communication about the interplay between work and home life is encouraged. The protocol is registered on the Open Science Framework (OSF) (https://doi.org/10.17605/OSF.IO/WFZU6).

**Funding:** The author(s) received no specific funding for this work.

**Competing interests:** The authors have declared that no competing interests exist.

## Author summary

It is becoming increasingly common for people to use smartphones for work purposes outside of their working hours. We looked at the published evidence and found that there was a relationship between the use of smartphone technology for work purposes in off-job hours and reported difficulties in maintaining boundaries between work and home life (referred to here as 'work-life conflict'). The strength of this relationship varied according to people's ability to 'switch off' from work, and whether they can openly talk to their managers about any impacts of work-related smartphone use (outside of their working hours) on their home lives. We suggest actions that managers can take to prevent or mitigate any potential negative impacts on digital technology during off-job hours on people's lives outside of work.

## Introduction

Worldwide, smartphone ownership and use has proliferated. The number of smartphone mobile network subscriptions reached almost 6.4 billion in 2022 and is forecast to exceed 7.7 billion by 2028 [1]. In the United Kingdom, the smartphone penetration rate has increased year-on-year and is anticipated to reach 92.4% by 2028 [2]. The use of smartphones is now ubiquitous, integrated into people's social and professional lives.

Smartphones go beyond older-design mobile phones by combining telephony with advanced computing capability, large storage capacity and Internet connectivity. In the context of work, smartphones have led to new ways of working, offering convenience in allowing staff to work flexibly from any location, resulting in faster real-time decision-making and the potential for increased workplace productivity [3]. However, their perceived impacts on productivity vary according to employment sector and job type [4]. Such digital devices can be utilised in diverse ways: communicating information, implementing workplace changes, offering a platform for health and wellbeing interventions [5,6], and/or providing a tool by which to promote autonomy, strengthen relationships with peers as well as superiors, and improve communication and knowledge-sharing [7,8]. The proposed benefits of mobile technologies, such as smartphones, are not limited to their use for work activity; it is suggested that using mobile technologies to engage in non-work activities during working hours (known as 'digital leisure') can, to some extent, contribute to employee overall well-being and productivity by means of mental recovery and replenishment [9].

While there are many benefits of the proliferation of smartphones, there are several caveats. Smartphone use in the workplace can lead to cyberloafing and cyberslacking (i.e., spending time on non-work-related digital activities at work) [10], distraction from work activities, and impaired work performance [11]. Some authors describe the 'dark side' of digital working including 'technostress', overload anxiety and addiction [12,13], resulting in lowered productivity both in the workplace and at home [14]. The continuous connectivity to the Internet afforded by smartphones, while offering flexibility to working adults [3], may lead to digital overuse, described as "a widespread social phenomenon sensitive to existing inequalities". [15]

Now that many work duties can be dealt with using smartphones in the home, there is a blurring of boundaries between work and non-work domains. According to Work-Family Border Theory [16], the likelihood of two domains (*viz.* work and family/home) with high permeability and flexibility to blend or integrate is high; thus, making an employee vulnerable to work-life conflict. Work-life conflict is a form of inter-role conflict that occurs due to role pressures derived from both home and work domains, which are perceived to be incompatible

or in conflict with one another [17]. Consequently, there are growing concerns about the immediate and long-term impact of the blurring of boundaries between work and home life on employees' work-life conflict [18].

However, there are individual differences in the impacts of mobile phones on the boundaries of work and home life, with some working adults perceiving their use during "off-job hours" to be more problematic than others [19,20]. Here, off-job hours are defined as work done, received, or happening away from or while not at one's job. Wright and colleagues [21] found that hours of work-related communication technology use outside of regular work hours can contribute to perceptions of work-life conflict, and that this predicted both job satisfaction and burnout. Further review evidence highlights the importance of addressing work-life conflict given its association with psychological, physical, and behavioural health [22]. The decreased segmentation between work and home resulting from smartphone use in off-job hours may, for some, lead to work-home interference, meaning pressures from work and home domains are mutually incompatible [23]. Indeed, the mere presence of a smartphone (in the knowledge of its constant connection to information) has been shown to reduce cognitive capacity and lead to smartphone-induced 'brain-drain', that is, where smartphones occupy most or all of our limited cognitive resources [24]. Conversely, other studies have highlighted the benefits of smartphone use during off-job hours; increasing opportunities for communication [25] and enhancing work flexibility as workers can bring their work tasks into the home domain [26]. Similarly, working mothers report smartphones increasing their sense of empowerment and interdependence when managing work and family commitments that, in turn, engenders a sense of work-life balance [27]. This refers to the "individual's perception that work and non-work activities are compatible and promote growth in accordance with an individual's current life priorities" [28], and contrasts to the perspective of conflict or interference between the work and personal domains by acknowledging the potential harmony between both domains.

In summary, studies of the influence of smartphone use during off-job hours present contradictory findings, highlighting both dysfunctional aspects (e.g., "usage patterns that are dangerous, distracting, anti-social and that infringe on work-life boundaries") and functional aspects (allowing users "to be efficient, to multitask without disruption to others, and to respond immediately to messages, as well as offering them the freedom to work from anywhere") [29]. While there are conceptual differences between the work-life balance and conflict, there is substantial inconsistency and overlap in how these terms are applied in research and practice [30]. Therefore, we elect to use "work-life conflict" as an all-encompassing term capturing both the conflict and opportunity between both work and life domains. There is a need to better understand the association between the use of smartphone technology for work purposes in off-job hours and the employees' work-life conflict, to inform recommendations for workers and their employers.

### Study aim

The aim of the study was to conduct a scoping review using a systematic approach to map relevant evidence examining the association between the use of smartphone technology for work purposes in off-job hours in relation to employees' self-reported work-life conflict.

## Materials and methods

A scoping review was the chosen method for reviewing the literature as it is well suited to rapidly developing areas of research. The protocol is registered on the Open Science Framework (OSF) (https://doi.org/10.17605/OSF.IO/WFZU6). The review was guided by Arksey and

O'Malley's [31] methodological framework, which has six stages including (i) identifying the research question; (ii) identifying relevant studies; (iii) study selection; (iv) charting the data; and (v) collating, summarising, and reporting the results, and (vi) stakeholder engagement. The review reporting aligns with the PRISMA-ScR (Preferred Reporting Items for Systematic reviews and Meta-Analyses extension for Scoping Reviews) checklist and explanation [32] (Supplementary file S1 Table).

## Stage 1: Identify the research question

Following an initial literature search, the research question we identified for this review was:

"What is the association between the use of smartphone technology for work purposes in off-job hours and employees' self-reported work-life conflict?".

The review objectives were: (i) to describe the extent, variety, and nature of the identified studies (including study focus, characteristics, and quality), (ii) synthesise findings (including identification of any mediators and moderators), and (iii) draw conclusions and identify gaps in the evidence to inform future research and practice.

## Stage 2: Identifying the relevant studies

The following databases were searched to identify applicable studies: PsycINFO, International Bibliography of the Social Sciences (IBSS), Academic Search Complete, ProQuest Central, Web of Science, ProQuest Theses, Emerald, Business Source Complete, ScienceDirect, and Scopus. Google Scholar was also searched for any additional articles that may not have been listed in the selected databases. An example search strategy for PsychINFO is available (S1 Text). Search terms and their free-text variants were identified in relation to two facets of the research question: smartphones (*"mobile devices" OR "mobile phone" OR "cell phone" OR "iPhone" OR "blackberry" OR "android phone" or "windows phone"*) and work-life conflict (*"work-family conflict" OR "work-life balance" OR "work-life interface" OR "work-home interference"*). Since Google Scholar does not have a "recent searches" option, which allows the combination of search queries to conduct an advanced search, we ran three searches; first, using the terms "smartphone" and "work-life conflict", second, using the terms "smartphone" and "work-home interference" and third, using the terms "smartphone" and "work-life balance". We reviewed the titles in the first five pages of each search followed by reviewing the abstracts and the full text against the inclusion/exclusion criteria. To identify additional relevant articles, reference lists of reviewed articles, and articles that cited included studies were searched.

Articles included in the study had to meet specific inclusion criteria covering four key domains: research methodology, study sample, specification of predictor and outcomes measure(s), and language restrictions. Specifically, we sought to identify studies that: sampled a working population aged 18 years or over, were published in English between 1st January 2012 and 29th November 2023 and quantified the relationship between the use of smartphone technology for work purposes during off-job hours and employees' experiences of work-home interference. Grey literature (including study protocols) was excluded. Studies were excluded from this review if the sample did not include working adults, articles were not in English, or the data were qualitative. To ensure that no study deviated from the overall aim of the current review, we reviewed operational definitions of variables under study and scrutinized scales, or measures used to quantify them. For instance, in the study by Schieman and Young [33], the variable "work contact" was operationalized as the degree to which participants sent or received email, phone calls, or text messages for work-related purposes during off-job hours.

Since two of these three tasks (*viz.* text messaging and making a phone call) are possible only on a mobile phone, the study was deemed appropriate for inclusion in the review.

## Stage 3: Study selection

The search strategy identified 1,104 potentially relevant studies: 1,097 articles from database searches and seven from reference list searches. One hundred seventy-two studies were duplicates and were removed, leaving 934 original studies to screen. The identified sources were reviewed using a two-stage review process. See Fig 1 for a flow diagram of the article selection process. At Stage one, titles and abstracts of identified sources (*n* = 934) were screened. Those

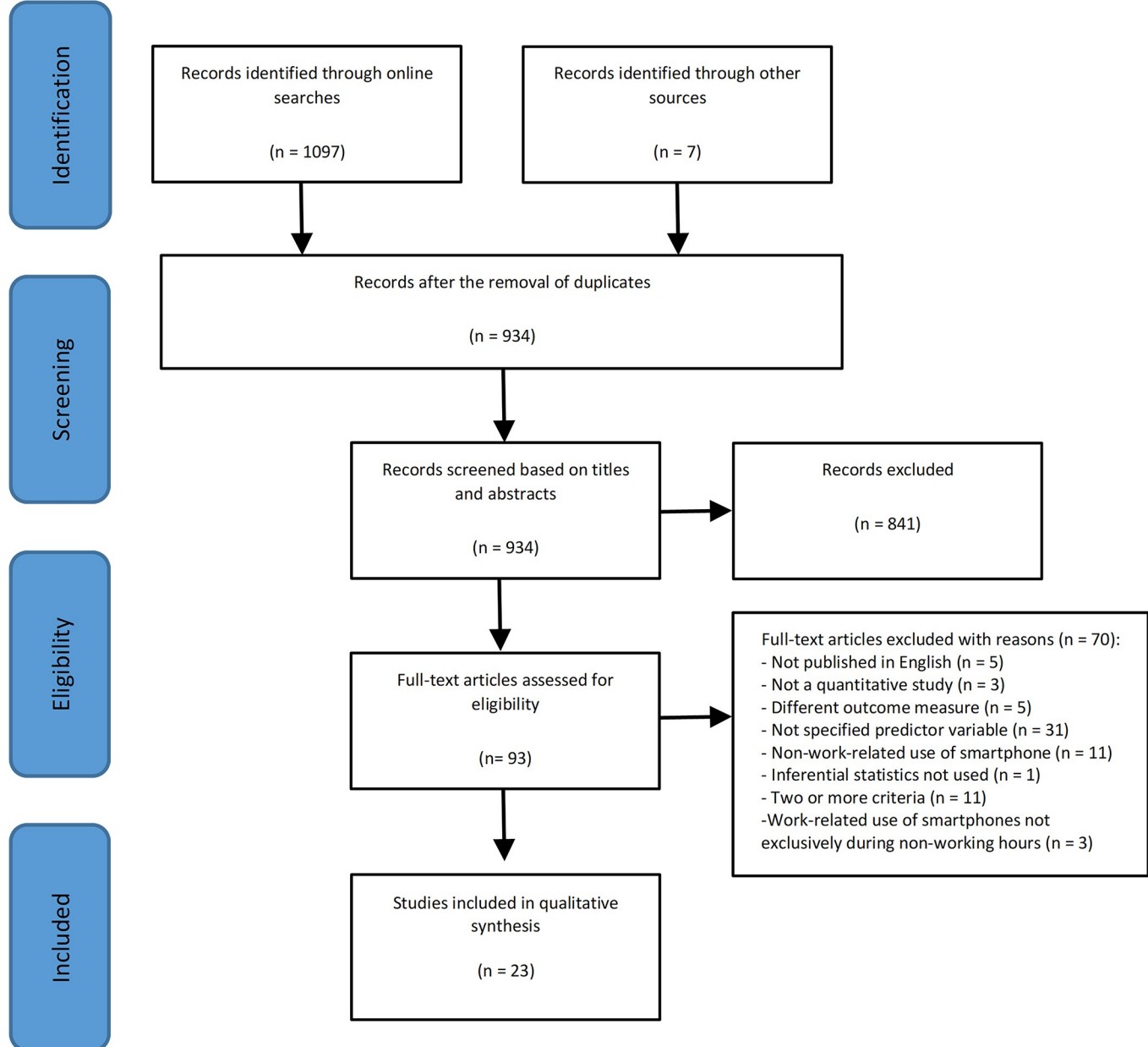

**Fig 1. The review process based on PRISMA flow diagram.**

studies that referred to work-related smartphone use and work-life conflict (or one of their related terms) were included in the full-text review stage. If it was unclear whether a study met inclusion criteria or not at the title and abstract stage, it was moved to the full-text review stage as a precautionary measure. In total, 841 studies were excluded at this stage leaving 93 articles to undergo a full-text review (Stage two). During this stage, all five specified inclusion criteria were applied. Seventy articles did not meet one or more of the specified inclusion criteria and were excluded. In total, 23 studies met all five specified inclusion criteria and were retained.

Review stages one (title and abstract screening) and two (full-text review) were carried out independently by one of the research team (JS). A random selection of 20% of articles at each stage were independently and blindly assessed by two other reviewers (JH and KT). The degree of inter-rater reliability was quantified using Cohen's Kappa. Strong inter-rater agreement was observed for both stages (stage one, $k$ = .83 [95% *CI* .67, .98] and .78 [95% *CI* .61, .95]; stage two, $k$ = 1 [95% *CI* 1, 1] and .86 [95% *CI* .67, 1.04]).

## Stage 4: Charting the data

A database was created in MS Excel and used to share articles between the reviewers, which facilitated data charting and consensus review. A data extraction form was developed as part of the research protocol to standardise the data extraction process. This form was peer-reviewed and piloted prior to its use. Data collected from each article included information related to the study's title, year of publication, authors' names, country, aim(s) of the study, theoretical framework(s) adopted to guide the investigation, hypotheses or research questions, predictors of work-life conflict, design, total sample size, response rate, percentages of male and female participants, other relevant details about the sample (e.g., industry, sector, designation etc.), scales used for measuring variables, and findings of the study.

## Stage 5: Collating, summarizing, and reporting the results

We used the framework approach described by Ritchie and Spencer [34], as used by Arksey and O'Malley [31]. This involved synthesising and interpreting the data by sifting and charting information based on the key themes identified in the literature. Thematic analysis was conducted by two researchers and any discrepancies in the analysis were resolved through discussion until consensus was reached.

For risk of bias (quality) assessment, a study quality assessment checklist was employed to examine the empirical rigour of included studies at study level, and to identify gaps in methodological practice. The quality assessment checklist was an adaptation of Caldwell et al.'s [35] framework of critiquing research. This checklist includes 26 items. The current study only utilised items relevant to quantitative research methods (19 items). A score was given for the presence of each criterion (2 = *fully met*, 1 = *partially met*, 0 = *not met* or *cannot tell)*; and then summed to give an overall rating for a study, with higher scores indicating strong methodological rigour.

## Stage 6: Stakeholder consultation

Stakeholder consultation is an optional stage in Arksey and O'Malley framework. The stakeholders were involved in Stage 1 (contributing to identifying the research question through knowledge of gaps in the literature and/or practice), Stage 5 (interpreting findings) and Stage 6 (considering the implications for practice and/or policy). The overall purpose of the inclusion of stakeholders was to assist in closing the gap between research production (i.e., the review findings) and research use (i.e., how our findings might be implemented in policy and practice). We were guided by design principles for engagement of stakeholders in research which

focus on three categories of principles: 'organisational', 'values' and 'practices' [36]. Stakeholders (*n* = 8) included employees and line managers (from micro-small, small, medium and large organisations) and organisational psychologists who were purposively identified through professional networks, and had a direct interest in the process and outcomes of this review. Their involvement was through virtual (video-conferencing or email) direct consultation with the research team to establish research priorities (Stage 1). They then reviewed and verified our interpretation of findings (Stage 5). Finally, they engaged in a brainstorming activity focused on knowledge translation to generate implications of the review findings for workplace policy and practice (Stage 6). This required minimal resources; approximately 2-hours of stakeholder time. At project end, the research team produced lay summaries for the stakeholders, of the scoping review and agreed research implications, to support organisational learning and reward stakeholder engagement.

## Results

### Overview

The review process yielded 23 studies for inclusion (Table 1 and S2 Table), that were conducted in the USA [37–44], the Netherlands [23,45–47], South Korea [48], Belgium [49], Canada [33,50], the UK [51], Malaysia [52], Sri Lanka [53], and South Africa [54–56]. One study [57] did not explicitly report the study location (although they recruited employees from a Scandinavian company). The publication year of studies ranged from 2012 to 2023, with six of the studies published in 2018.

### Study designs and settings

Across the included studies, a variety of research designs were employed: diary-entry (n = 4) [23,45–47], repeated measures (n = 1) [38], cross-sectional (n = 17) [33,37,39–44,48–56] and longitudinal (n = 1) [57] designs. Data in all the studies were collected using convenience sampling. Except for seven studies that specifically recruited employees from the construction [42,54–56], telecommunications [51,57], or accounting [50] sectors, participants in other studies were recruited across sectors. Samples across studies was heterogenous in terms of participants' designations or job roles.

### Study focus

Of the 23 included studies, 13 [23,37,38,41–43,45–47,49,51,52,57] clearly operationalised and measured the impact of work-related smartphone use during off-job hours. Two studies [39,50] examined the use of mobile devices (a smartphone or an internet-enabled tablet) for work-related purposes during off-job hours; one study [48] explored the impact of distinct attributes of smartphone use for work (namely, work overload, autonomy, flexibility, and productivity); and seven studies [33,40,44,53–56] examined the impact of work-related use of information and communication technology (ICT) devices outside working hours. The operationalisation and measurement of smartphone technology and work-life conflict in included studies, and additional study variables, are shown in Supplementary file S2 Table.

### Operationalisation of smartphone use

When reviewing the operationalisation of smartphone use there was a variety of conceptual and measurement approaches. Having reviewed the 23 included studies, we have therefore categorised the operationalisation of their independent variable into two thematic areas. First, the structural use of smartphone technology, which we define as the *functional use* (e.g., time

**Table 1. Characteristics of Included Studies.**

| Study | Study aim(s) | Sample (% males, % females, response rate, and other relevant details) | Study design (country, study population, and other relevant details) | Theoretical framework(s) |
|---|---|---|---|---|
| Brown and Palvia [37] | To explore relationships among work-related mobile device usage while at work, work-related mobile device usage while at home, personal mobile device usage at home, productivity, employer expectations, flexibility of work structure, and work-life conflict. | N = 165 (55%, 45%, 58%, and majority of participants [31%] were mid-level managers). | Cross-sectional design (USA, employed and smartphone users). | Work/family border theory [16] |
| Derks et al. [45] | To examine the impact of smartphone-use for work-related activities during non-working hours on recovery strategies (psychological detachment, relaxation, mastery, and control activities) adopted by employees. | N = 80 (78%, 22%, N/A, $n_1$ [smartphone group] = 40, $n_2$ [control PC-group] = 40, participants were employed in 22 different organisations but, were similar in their workload and job type). | Diary-entry design with control group (The Netherlands, employed and smartphone users). Participants were contacted via email for 6 workdays over a period of 2 weeks. | Effort-Recovery Theory [58] |
| Derks et al. [46] | The aims of the study were threefold. First, to examine the moderating role of segmentation preference in the relationship between daily work-related smartphone use during off-job hours and daily work-family conflict (WFC). Second, to investigate the moderating role of segmentation preference in the relationship between daily work-related smartphone use during off-job hours and daily family role performance. Third, to examine the mediating role of WFC in the moderated relationship (by segmentation preference) between daily work-related smartphone use during off-job hours and daily family role performance. | N = 71 (56%, 44%, N/A, participants worked in diverse fields, 60% of the participants had a university degree, 63% of participants were living with a partner, and 37% had children living at home). | Diary entry design (The Netherlands, smartphone users who worked at least 4 days a week). Participants were contacted via email for 4 successive workdays within one working week. | Boundary theory [16,59,60] |
| Ragsdale and Hoover [38] | To examine the impact of work-related cell phone use during non-working hours on (i) emotional exhaustion, (ii) work engagement, and (iii) work-family conflict, and to explore the moderating role of cell phone attachment in these relationships. | N = 313 (48%, 52%, 28%, participants were adults, had a full-time job, owned a cell phone, and worked in diverse fields, and majority of the participants were married or cohabiting, had children, and held a university degree). | Repeated-measures design (USA, employed full time and smartphone users). Work-related cell phone use and cell phone attachment were assessed at time 1 (T1), and emotional exhaustion, wok engagement, and work-family conflict were assessed at time 2 (T2). The time gap between the two surveys was one week. | Job Demands-Resources Model [61] |
| Derks et al. [47] | To examine the impact of daily smartphone use for work-related purposes during after work hours on daily work-home interference, and to explore the moderating role of supervisor expectations, social norms set by colleagues, and daily work engagement in these relationships. | N = 100 (75%, 25%, N/A, 85% of participants lived with a partner, 67% of participants had children living at home, 72% of participants held a university degree, and participants worked in a diverse range of white-collar sectors). | Diary-entry design (The Netherlands, employed full time, organisation provided smartphone users). Participants were contacted via email for 4 successive workdays within one working week. | Boundary theory [16,59], Equity theory [62,63], and Social Learning theory [64] |

*(Continued)*

**Table 1.** (*Continued*)

| Study | Study aim(s) | Sample (% males, % females, response rate, and other relevant details) | Study design (country, study population, and other relevant details) | Theoretical framework(s) |
|-------|--------------|------------------------------------------------------------------------|----------------------------------------------------------------------|--------------------------|
| Derks and Bakker [23] | The aims of the study were sixfold. First, to examine the negative impact of daily recovery (psychological detachment and relaxation) on daily work-home interference (WHI). Second, to investigate the positive relationship between daily WHI and daily burnout symptoms (exhaustion and cynicism). Third, to examine the mediating role of reduced daily WHI in the negative relationship between daily recovery and daily burnout symptoms. Fourth, to examine the positive relationship between work-related smartphone during non-working hours and daily WHI. Fifth, to investigate the moderating role of intensive smartphone use in the negative relationship between daily recovery and daily WHI. Sixth, to examine the moderating role of smartphone use in the positive relationship between daily WHI and daily burnout symptoms. | N = 69 (31.9%, 68.1%, N/A). Majority of the participants (71%) were "highly educated" (p. 420; the level of education [undergraduate or postgraduate degree] was not specified). | Diary-entry design (The Netherlands, full-time employees using a company-provided smartphone). Participants were contacted via email for 5 successive workdays in a working week. | Effort-Recovery theory [58] |
| Carlson et al. [39] | To examine the impact of work-related mobile device use during family time by job incumbents on their work-to-family conflict (WFC) and the impact of job incumbents' WFC on spouses' family-to-work conflict (FWC), job satisfaction, and job performance via relationship tensions between job incumbents and spouses. | N = 344 pair (job incumbents– 61%, 39%, N/A; spouses– 39%, 61%, N/A, couples were married for an average of 13 years, 68% of couples had children living at home). | Matched-pairs, cross-sectional design (USA, married, full-time employees who used a mobile device for work and non-work purposes). | Work-family crossover model [65], Family Systems Theory [66], and Work-home resources model [67] |
| Yun et al. [48] | To explore the impact of the attributes of office-home smartphone (OHS; work overload, flexibility, autonomy, and productivity) on employees' levels of work-life conflict, stress, and user resistance to OHS. In addition, to examine the impact of segmentation culture on work-life conflict. | N = 300 (65%, 35%, 40%, majority of the participants were single [54%], did not have children [62%], and worked in manufacturing or sales [31%]). | Cross-sectional design (South Korea, smartphone users). | Role boundary theory [59] |
| Ferguson et al. [40] | To explore the impact of mWork on job incumbent's turnover intentions via two pathways: (i) mWork leading to work-family conflict, which further leads to burnout and reduced organizational commitment, and (ii) mWork leading to work-family conflict for job incumbent, which further leads to spousal resentment towards the incumbent's organisation and reduced commitment towards the incumbent's organisation. | N = 344 pairs (job incumbents– 39%, 61%, NR; spouses– 61%, 39%, NR, couples were married for an average of 13 years and 68% of couples had children living at home. The sample was heterogenous in terms of industry/sectors, and salary scales). | Matched-pairs, cross-sectional design (USA, married, employed full time, and mobile device users). | Conservation of resources theory [68], and Family Systems Theory [69] |

(*Continued*)

**Table 1.** (Continued)

| Study | Study aim(s) | Sample (% males, % females, response rate, and other relevant details) | Study design (country, study population, and other relevant details) | Theoretical framework(s) |
|---|---|---|---|---|
| Gadeyne et al. [49] | To examine the moderating roles of integration preferences, organizational integration norms, and work demands in the relationship between work-related use of information and communication technological (ICT) devices (smartphones and PCs/laptops) and work-to-home conflict. | N = 467 (15%, 85%, N/A, majority of the participants [92%] were cohabiting with partners and working as clerks [52%]. Participants had an average of two children living in their households. | Cross-sectional design (Belgium, employed parents with at least one child under the age of 12 years, smartphone users). | NR |
| Schieman and Young [33] | To examine the impact of work contact on work-to-family conflict, and to investigate the moderating roles of job pressures and job resources (job autonomy, some/full schedule control, and challenging work) in these relationships. | N = 5729 (52%, 48%, 40%, 48% of participants were married or living with a partner, and 40% had children younger than 18 years of age living in the household). | Cross-sectional design (Canada, employed, and live in non-institutional residence). | Border theory [16,70], and Job Demands-Resources model [61] |
| Harris [41] | The aims of the study were threefold. First, to examine the impact of work-life balance on stress, life satisfaction, and job satisfaction. Second, to examine the impact of smartphone intrusion on work-life balance. Third, to explore the moderating role of organisation's attitude towards smartphone use in these relationships. | N = 202 (57.1%, 41.9%, N/A, 35% of participants reported having a company-provided smartphone). | Cross-sectional design (USA, paid employees, smartphone users). | Spillover Theory [71] |
| Burney [42] | To explore the effects of personal smartphones, company-sponsored smartphones, and both on levels of work-life balance of managerial employees in the property construction industry. | N = 162 (11.73%, 88.27%, N/A, 54.32% of participants were married, 65.43% had children living at home, 32.1% of participants used personal smartphones for work, 23.46% of participants used company-issued smartphones for work, and 44.44% of participants used both for work). | Sequential explanatory mixed-methods design (USA, managers in property management, smartphone users [personal, company, or both]). | Work-Family Border Theory [16] and Spillover Theory [72] |
| Ward and Steptoe-Warren [51] | To explore the impact of using BlackBerry (BB) devices for work-related purposes during non-working hours on employee's work-family conflict and wellbeing; and to examine job control and psychological detachment from work as mediators. | N = 86 (75.6%, 24.4%, 39.13%, 61.63% of participants were senior managers, and 38.37% of participants were junior managers). | Cross-sectional design (UK, employed in a leading communications service company, possessed a company-issued BB device for work purposes). | Conservation of Resources Theory [73] |
| Wei and Teng [52] | To study the impact of work-related smartphone outside of official working hours on work-life conflict and work engagement, and to examine the moderating role of the employment sector (public vs. private) in these relationships. | N = 229 (42.4%, 57.6%, N/A, majority of the participants had an undergraduate degree [69.4%], held managerial positions [53.3%], and worked in private sector [72.1%]). | Cross-sectional design (Malaysia, employed, smartphone users). | NR |
| Bowen and Zhang [54] | The aims of the study were threefold. First, to examine the antecedents and consequences of work-family conflict (WFC). Second, to explore the role of cross-boundary work contact on WFC. Third, to investigate the inter-relationships between WFC and family-work conflict (FWC). | N = 690 (81%, 19%, N/A, 35% of participants were architects). | Cross-sectional design (South Africa, employed construction professionals). | Job Demands-Resources model [74], and Boundary theory [59,75] |

(*Continued*)

**Table 1.** (Continued)

| Study | Study aim(s) | Sample (% males, % females, response rate, and other relevant details) | Study design (country, study population, and other relevant details) | Theoretical framework(s) |
|---|---|---|---|---|
| van Zoonen et al. [57] | To examine the mediating impact of boundary spanning communication on the relationship between work-related smartphone use during non-working hours, and work-life conflict and organisational identification. | N = 367 (54.9%, 45.1%, 54.4% [T1], 49.3% [T2], 32.7% of participants had a university degree, 37.6% graduated from an applied university, and 53% of participants had at least one child living at home). | Longitudinal design (NR, knowledge workers in a large Scandinavian telecommunications company, smartphone users, time gap between two administrations was 1 year–the first survey measured employees' work-related smartphone use after hours and the second survey measured boundary spanning communication, work-life conflict, and organisational identification). | Boundary theory [59], Work-family border theory [16], and Structurational perspective on identification [76] |
| Bowen et al. [55] | To examine the construct validity and internal consistency of modified versions of scales originally developed by Schieman and Young (33) to assess smartphone use (work contact), work-family conflict, working conditions, psychological distress, and sleep problems. | N = 630 (82%, 18%, N/A, 88% of participants were married or living with a partner, 49% of participants had children living at home, and 58% of participants were partners or directors). | Cross-sectional design (South Africa, employed construction professionals). | NR |
| Bowen et al. [56] | To explore the impact of work contact (including, using a smartphone technology in non-working hours) and work–family conflict on psychological distress and sleep problems. | N = 630 (82%, 18%, N/A, 88% of participants were married or living with a partner, 49% of participants had children living at home, and 58% of participants were partners or directors). | Cross-sectional design (South Africa, employed construction professionals). | Job Demands-Resources model [74], and Boundary theory [59,75] |
| Fender [43]* | The aims of the study were multifold*. Firstly, to examine the moderating role of after-hours electronic communication (AEC) expectations in the relationship between work extending communication (WEC), and receptive electronic communication (REC) behaviour and electronic tethering (ET). Secondly, to examine the positive relationship between REC behaviours and ET. Thirdly, to examine the positive relationship between REC behaviours and work-to-family conflict. Fourthly, to examine the moderating role of work-to-home segmentation preferences in the relationship between work-to-family conflict, and psychological and physiological strain, job satisfaction and affective organizational commitment. Fifthly, to investigate the moderating role of ET instrumentality in the relationship between ET, and psychological and physiological strain, job satisfaction and organizational commitment. | N = 285 (57%, 43%, NA, 45% of participants had an undergraduate degree, and 61% of them had a managerial role). | Cross-sectional design (USA; employees with cell/smartphones that organizations could use to contact them). | Role Theory [77]; Field theory of unfreezing-movement-refreezing [78]; General Adaptation Syndrome [79]; Transactional theory of stress [80]; Job Demands-Control model [81]; Control model of stress [82]; Person-Environment Fit model [83]; Conservation of Resources Theory [73] |

(*Continued*)

**Table 1.** (Continued)

| Study | Study aim(s) | Sample (% males, % females, response rate, and other relevant details) | Study design (country, study population, and other relevant details) | Theoretical framework(s) |
|---|---|---|---|---|
| Mansour et al. [50] | The aims of the study were threefold. Firstly, to examine the positive relationship between work intensification and use of smartphone and/or tablet for business purposes outside working hours. Secondly, to examine the relationship between the use of smartphone and/or tablet for business purposes outside working hours and work-family conflict (WFC). Thirdly, to examine the mediating role of the use of smartphone and/or tablet for business purposes outside working hours in the relationship between work intensification and WFC. | N = 388 (33%, 67%, NR, 33.2% of participants had 11–20 years of work experience, 76.8% of participants lived with their partner and children, 45.9% of participants had 2 children, 61.1% of participants worked in the private sector, and 39.8% of participants had a senior management position). | Cross-sectional design (Quebec Province, Canada, accounting professionals who lived with children). | Conservation of resources theory [84]; Job demands-resources model [74,85,86] |
| Alwis and Hernvall [53] | The aims of the study were: (i) to examine the impact of segmentation preference on perceived intensity of information and communication technologies (ICTs) at work and work-life conflict, and; (ii) to examine the mediating role of perceived intensity of ICTs at work in the relation between segmentation preference and work-life conflict. | N = 225 (52.9%, 47.1%, 23%, 55.6% of participants were married, 59.6% had a child living at home, 68.9% had elderly dependents at home, and 48.5% had an executive position). | Cross-sectional design (Sri Lanka, employees working in a diverse range of industries were recruited). | Boundary theory [87] |
| Moore [44] | The aim of the study was to examine the association between after-hours communication (cell phone and computer exchange and Facebook use), and work-life balance and job satisfaction. | N = 153 (24.2%, 75.2%, NR). | Cross-sectional design (USA, participants worked in a diverse range of industries). | Not mentioned |

*The study by Fender [43] examined ten hypotheses. Due to practical reasons, hypotheses related to this review are mentioned in the table. For a more details, readers are directed to the section, "CHAPTER 3 –RESEARCH MODEL AND HYPOTHESES" (p. 65) in Fender [43].

spent answering work emails) of this form of technology to conduct work-related tasks in off-job hours. Second, the psychosocial use of smartphones for work purpose, which we define to be *perceptual use*, relating to employees' feelings, emotions, or perceptions regarding using a smartphone for work related purposes during off job hours (e.g., pressure to respond to work emails during off job hours). Using this categorisation system, we observed that nine studies [33,39,40,49,51,54–57] included in this review examined functional use, six studies [23,44,48,50,52,53] examined perceptual use, and five studies [38,41,43,46,47] investigated both functional and perceptual use. In the case of three studies [37,42,45], it was not clear whether they assessed functional or perceptual aspects of work-related smartphone use in off-job hours. It is important to highlight here that among the five studies that measured both the perceptual and functional aspects, only two studies [41,43] distinguished between the two.

## Operationalisation of work-life conflict

For work-life conflict, the most frequently examined outcome was work-family conflict (n = 11) [33,38–40,43,46,50,51,54–56], followed by work-home interference (n = 3) [23,45,47], work-life balance (n = 3) [41,42,44], work-life conflict (n = 5) [37,48,52,53,57], and work-to-home conflict (n = 1) [49]. Regarding measurement of work-life conflict, except for one study[11], other studies used standardised scales with established psychometric properties. The most frequently used measure to quantify work-life conflict was the scale developed by Carlson et al. [88], followed by the SWING scale [89].

## Risk of bias quality assessment

The results of quality assessment are presented in Table 2 and reflected on in the discussion. Most studies were homogeneous in terms of their methodological quality (total score range: 21–36, M = 29.87, SD = 4.15). The least commonly met or partially met criteria included: the identification of ethical issues and how these were addressed, identification of the research methodology and its justification, and identification of and rationale behind the adopted research design.

## Key themes

Three key themes were identified using principles of framework analysis [34] involving synthesis of findings: (i) Relationship between Work-Related Smartphone Use During Off-Job Hours and Work-Life Conflict, (ii) Mediators and Moderators of the Relationship between Work-Related Smartphone Use During Off-Job Hours and Work-Life Conflict, and (iii)

**Table 2. Evaluation of Included Studies Using a Study Quality Checklist.**

| Quality Assessment Criteria | Study Number | | | | | | | | | | | | | | | | | | | | | | |
|---|---|---|---|---|---|---|---|---|---|---|---|---|---|---|---|---|---|---|---|---|---|---|---|
| | 37 | 45 | 46 | 38 | 47 | 23 | 39 | 48 | 40 | 49 | 33 | 41 | 42 | 51 | 52 | 54 | 57 | 55 | 56 | 43 | 50 | 53 | 44 |
| Title reflects content | 2 | 2 | 2 | 1 | 2 | 2 | 2 | 1 | 2 | 2 | 2 | 1 | 2 | 2 | 2 | 2 | 2 | 2 | 2 | 2 | 1 | 2 | 2 |
| Authors credible | 2 | 2 | 2 | 2 | 2 | 2 | 2 | 1 | 2 | 2 | 2 | 2 | 2 | 2 | 2 | 2 | 2 | 2 | 2 | 2 | 2 | 2 | 2 |
| Abstract summarises the key components of the study | 2 | 2 | 2 | 2 | 2 | 2 | 2 | 2 | 2 | 2 | 2 | 1 | 1 | 2 | 2 | 2 | 2 | 2 | 2 | 2 | 2 | 1 | 1 |
| Rationale for research clearly outlined | 2 | 2 | 2 | 2 | 2 | 2 | 2 | 2 | 2 | 2 | 2 | 2 | 2 | 2 | 2 | 2 | 2 | 2 | 2 | 2 | 1 | 2 | 1 |
| Literature review is comprehensive and up to date | 2 | 2 | 2 | 2 | 2 | 2 | 2 | 2 | 2 | 2 | 2 | 2 | 2 | 1 | 2 | 2 | 2 | 2 | 2 | 2 | 1 | 2 | 1 |
| Aim of the study clearly stated | 2 | 2 | 2 | 2 | 2 | 2 | 2 | 2 | 2 | 2 | 2 | 2 | 2 | 2 | 2 | 2 | 1 | 2 | 2 | 1 | 2 | 1 | 2 |
| Ethical issues identified and addressed | 1 | 0 | 1 | 0 | 2 | 0 | 2 | 0 | 0 | 0 | 0 | 0 | 2 | 2 | 1 | 2 | 0 | 1 | 0 | 0 | 0 | 0 | 0 |
| Methodology identified and justified | 0 | 1 | 1 | 1 | 1 | 1 | 0 | 0 | 0 | 0 | 0 | 0 | 0 | 2 | 0 | 0 | 0 | 0 | 0 | 0 | 0 | 0 | 2 |
| Study design is clearly identified and rationale for choice of design evident | 1 | 2 | 2 | 1 | 2 | 2 | 1 | 1 | 1 | 1 | 1 | 0 | 0 | 1 | 1 | 1 | 1 | 2 | 1 | 2 | 0 | 1 | 0 |
| Study hypothesis stated and key variables clearly defined | 2 | 2 | 2 | 2 | 2 | 2 | 2 | 2 | 2 | 2 | 0 | 2 | 2 | 2 | 2 | 2 | 2 | 2 | 0 | 2 | 2 | 1 | 1 |
| Population clearly defined | 2 | 2 | 2 | 2 | 2 | 2 | 2 | 1 | 2 | 2 | 2 | 2 | 2 | 0 | 2 | 2 | 2 | 0 | 1 | 1 | 2 | 1 | 0 |
| Sample is adequately described and reflective of the population | 2 | 1 | 2 | 0 | 2 | 2 | 2 | 1 | 2 | 1 | 2 | 2 | 2 | 0 | 0 | 2 | 1 | 1 | 2 | 1 | 2 | 2 | 2 |
| Is there a control group? Are samples matched? | 2 | 1 | 2 | 2 | 2 | 1 | 1 | 2 | 2 | 2 | 2 | 1 | 2 | 1 | 1 | 2 | 2 | 2 | 2 | 0 | 0 | 0 | 0 |
| Method of data collection valid and reliable | 2 | 2 | 2 | 1 | 2 | 2 | 2 | 2 | 2 | 2 | 2 | 2 | 2 | 2 | 2 | 2 | 2 | 2 | 1 | 1 | 2 | 2 | 2 |
| Method of data analysis valid and reliable | 2 | 2 | 2 | 2 | 2 | 2 | 2 | 2 | 2 | 2 | 2 | 1 | 2 | 2 | 1 | 2 | 2 | 2 | 2 | 1 | 2 | 2 | 1 |
| Results presented in an appropriate and clear manner | 2 | 2 | 2 | 2 | 2 | 2 | 2 | 2 | 2 | 2 | 2 | 2 | 1 | 2 | 2 | 2 | 2 | 2 | 2 | 1 | 2 | 2 | 1 |
| Discussion is comprehensive | 2 | 2 | 2 | 2 | 2 | 2 | 2 | 2 | 2 | 2 | 2 | 2 | 2 | 2 | 2 | 2 | 2 | 2 | 2 | 2 | 1 | 2 | 1 |
| Results are generalizable | 2 | 1 | 1 | 1 | 1 | 1 | 2 | 2 | 2 | 2 | 2 | 1 | 1 | 1 | 1 | 2 | 2 | 2 | 2 | 1 | 1 | 1 | 1 |
| Conclusion is comprehensive | 2 | 2 | 2 | 1 | 2 | 2 | 2 | 2 | 2 | 2 | 2 | 1 | 2 | 2 | 1 | 2 | 2 | 2 | 2 | 2 | 1 | 2 | 1 |
| **Total Score** | **22** | **32** | **35** | **28** | **36** | **33** | **34** | **29** | **33** | **32** | **31** | **26** | **33** | **29** | **27** | **35** | **32** | **32** | **29** | **28** | **24** | **26** | **21** |

Relationship between Work-Related Smartphone Use During Off-Job Hours and Workers' Wellbeing, Attitudes and Behaviours. Themes (i) and (ii) directly relate to the research question and objectives (i) and (ii). The third theme relates to objective (iii) and was identified following synthesis of findings from the review, and highlights the diversity of outcome measures in the included studies.

(i) Relationship between Work-Related Smartphone Use During Off-Job Hours and Work-Life Conflict

Of the 23 studies, 19 [23,33,37–44,46,47,50–56] observed a significant association between increased use of smartphone for work purposes in off-job hours and increased work-life conflict (Supplementary file S3 Table). A comparison of effect sizes with regards to the operationalisation of work-related smartphone use in off-job time (functional, perceptual, both, or unclear; Table 1) and the quality of included studies (Table 2) revealed that there was little difference in the degree of the relationship observed (i.e., the effect sizes across studies ranged from small-to-moderate; Supplementary file S3 Table). Studies that did not observe a statistically significant finding did not notably differ with regards study quality, sample size or other study characteristics.

(ii) Mediators and Moderators of Work-Related Smartphone Use During Off-Job Hours and Work-Life Conflict:

When reviewing these 23 studies, we observed that a large proportion investigated a wider variety of dependent variables beyond work-life conflict. A key finding from this review is the variety of variables that have been tested and explored in seeking to understand the postulated association between work-related smartphone use during off-job hours and work type conflict. Many of the included studies explored the contributory role of potential moderators or mediators within this association [33,38,41,46,47,49,51,52,57]. An overview of those studies that tested the role of a third variable as a potential mediator or moderator within the association between work related smartphone use during off-job hours and self-reported work-life conflict is provided (Supplementary file S4 Table).

The mediators identified in our sample of studies included: psychological detachment from work (i.e. detachment from work, when not at work)[14] and communication about family demands with one's supervisor [57]. Specifically, the frequency and duration of BlackBerry usage outside of working hours was negatively associated with psychological detachment, which was further negatively associated with work-family conflict [51]. Regarding the second mediator, smartphone use after working hours was positively associated with communication about family demands with supervisor, which was further negatively associated with work-life conflict [57]. Job control (i.e., a person's ability to influence what happens in their work environment) [51] and communication about work demands with one's family members [57] did not appear to significantly mediate the relationship between smartphone use and work-life conflict. Moderators found to strengthen the relationship between work-related smartphone use in off-job time and work-life conflict included: supervisor expectations [47] and job pressure [33]. Moderators found to attenuate the strength of the relationship included: low segmentation preference [46] (i.e., the degree to which one prefers to separate various aspects of work and family from each other by creating boundaries around the work and family domains), cell phone attachment [38], daily work engagement (i.e., the degree of personal investment in one's work role) [47], job autonomy (i.e., the degree to which one has control over *how* to get the job done) [33], full schedule control (i.e., the degree to which one has control over *when* and *where* to get the job done [33], challenging work [33], and organisation's attitude towards smartphone use [41]. Variables that were not found to moderate the relationship included: norms set by colleagues [47], integration preference (i.e., preference for how one coordinates their personal and professional lives in a complementary way and fulfills both

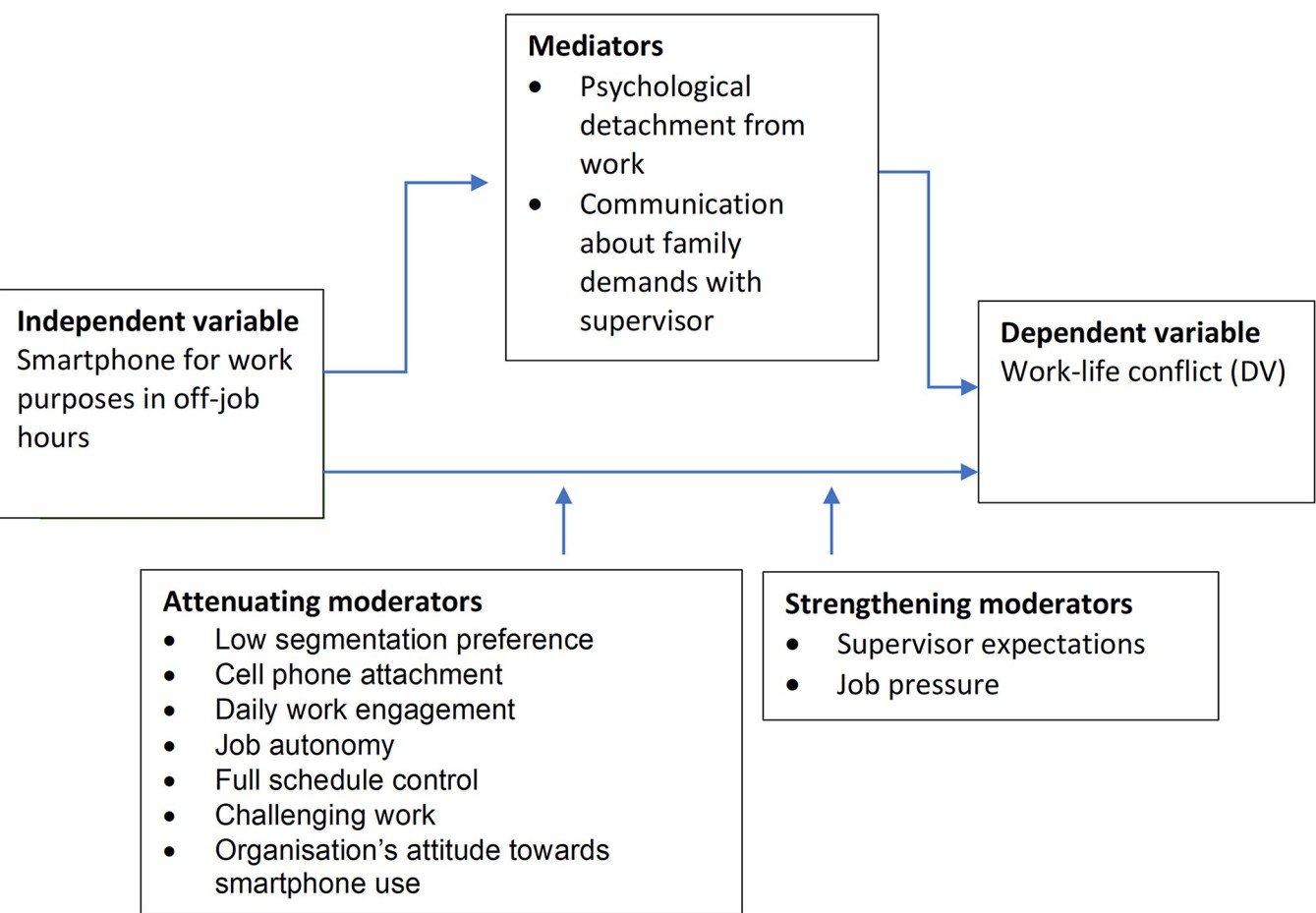

**Fig 2. Mediators and moderators of the relationship between smartphone use for work purposes in off-job hours and work-life conflict.**

sets of responsibilities) [49], integration norms (i.e., norms observable within the organisation for how other coordinate their personal and professional lives in a complementary way and fulfill both sets of responsibilities) [49], work demands [49], and some schedule control [33]. See Fig 2 for key mediators and moderators.

(iii) Relationship between Work-Related Smartphone Use During Off-Job Hours and Workers' Wellbeing, Attitudes and Behaviours.

In addition to work-life conflict, included studies examined the association between work-related smartphone use during off-job hours and several aspects of employees' wellbeing (both negative and positive aspects), attitudes, and behaviours (Supplementary file S5 Table). Regarding negative aspects of wellbeing, three studies reported low to moderate positive associations between work-related smartphone use during off-job hours, and measures of job stress [41], psychological distress [33], psychological and physiological strain [43], and sleep problems [56]. In one study, job autonomy and challenging work attenuated the relationship between increased work-related use of smartphone in off-job hours and sleep problems; whereas, in contrast, job pressure amplified this observed association [33], albeit to a minimal extent. The use of smartphones to attend to work-related matters during nonworking hours hindered engagement in recovery activities (such as, relaxation, mastery, and control/autonomy) [45], and fostered the intrusion of personal life into work life and vice-versa [41].

Unexpectedly, the association with positive aspects of well-being (such as, life satisfaction [41], job satisfaction [41,44], and work engagement [38,52]) was similar in degree and direction to the association between work-related smartphone use during off-job hours and the negative aspects of wellbeing mentioned above. This challenges the assumption that positive and negative aspects of wellbeing are at opposite ends of a spectrum. In one study, the frequency of smartphone use for work during personal time was associated with increased life satisfaction and job satisfaction [41] albeit weakly. However, in the same study, increased perceived work life to personal life smartphone intrusion was associated with decreased job satisfaction [41]. A positive, but weak, relationship between smartphone use for work-related purposes during off-job time and work engagement was found in two included studies [38,52], in contrast to what might be expected in the wider literature. This relationship was positively moderated by employees' cell phone attachment in one study [38].

Regarding employees' attitudes towards their work or job, work-related smartphone use during off-job time was found to promote affective organisational commitment [43] and organisational identification [57]. The relationship with the latter was partly mediated by communication about family demands with supervisors [57]. Lastly, regarding employee behaviour, work-related smartphone use during off-job time was found to enhance job performance [43] and family role performance [46]. Work-related smartphone use during off-job time reinforced communication about family demands with a supervisor, as well as communication about work demands with family members [57].

## Discussion

To our knowledge, this is the first scoping review to map the published evidence examining the association between the use of smartphone technology for work purposes in off-job hours in relation to employees' self-reported work-life conflict. In doing so, we also unpack potential mediators and moderators of this relationship, as well as related outcomes of off-job hours smartphone technology use in relation to worker wellbeing, attitudes, and behaviours.

Overall, most of the studies identified a significant association between increased use of smartphone for work purposes in off job-hours and increased work-life conflict with small-to-moderate effect sizes. They highlight the heterogenous manner in which home and life domains are considered, including its focus (e.g., family vs home life) and the nature of the overlap between both domains (i.e., where they interfere or harmonise) [30]. Additionally, the included studies highlight a negative psychological and behavioural impact on employees of increased use of smartphone for work purposes in off job-hours, including job stress and strain, and sleep disturbances. As such, our review findings emphasise the 'dysfunctional aspect' of smartphone use during off-job hours (infringement on work-life boundaries) as described by Middleton and Cukier [29] and lend support to the Work-family Border Theory [16]. This theory purports the vulnerability of individuals to work-life conflict due to the high likelihood of work and family/home lives integrating. Having an awareness of the strong association between use of smartphone for work purposes in off job-hours and work-life conflict is important, both to employees and employers, since work-life conflict has been shown to predict job satisfaction and burnout [21] both of which, in turn, predict turnover intentions [90].

In this review, we found that the relationship between use of smartphone for work purposes in off job-hours and work-life conflict was mediated by psychological detachment from work, and communication about family demands with one's supervisor. The first key mediator, psychological detachment, specifically in the digital era (i.e., the creation of boundaries around information and communication technology), has been associated with lower levels of work presenteeism and higher levels of family-life satisfaction [91]. The second key mediator

highlights the important role of the line manager (and employee communication with them) in this process. It is well established that managers contribute to the development of policy relating to work-life balance, and play a pivotal role in translating work-life balance policies into practice [92]. Drawing on Boundary Theory [59], smartphone use can make boundaries between work and life more permeable, and employees may need to communicate any concerns relating to this to their line managers to reduce work-life conflict. Such discourse between the employee and their manager(s) relies on organisations establishing a psychologically safe work environment, in which employees feel safe to speak up about concerns (e.g., the impact of work connectivity in off-job hours on family life). Studies have demonstrated that psychological safety in the workplace is an important predecessor for interpersonal communication [93]. Having open conversations with line managers about after-hours connectivity may help employees to establish clear expectations, reduce stressors associated with connectivity, and ultimately reduce work-life conflict [57].

This review identified key moderators of the relationship between increased use of smartphone for work purposes in off-job hours and increased work-life conflict. Moderators that strengthened this relationship were supervisor expectations and job pressure. High after-hours availability expectations (i.e., from managers / supervisors) has been associated with low psychological detachment from work, and it has been recommended that the introduction of 'availability' policies and discouragement of work-related smartphone use outside regular work hours may help employees to achieve successful boundary control and subsequent psychological detachment [94]. This is important given the known relationship between psychological detachment, workload (i.e., job pressure) and wellbeing (e.g., Sonnentag and Bayer [95]).

In our included studies, moderators that attenuated the strength of the relationship between use of smartphone for work purposes in off-job hours and work-life conflict include low segmentation preference, cell phone attachment, daily work engagement, job autonomy, full schedule control, challenging work, and organisation's attitude towards smartphone use. Low segmentation preference refers to the tendencies of individuals to separate their working and non-working roles. Employees with higher segmentation appeared to have less problems (e.g., work-life conflict) caused by work connectivity behaviour using smartphones in off-job hours [46]. Other research found that segmentation norms of the team moderate the relationship between work-family segmentation preferences and work-related ICT use at home [96], although norms within the organisation (i.e., integration norms / norms set by colleagues) were not found to be significant moderators in the studies included in this review [47,49]. Cell phone attachment (i.e., valuing and being physically attached to a cell phone) has been found to buffer the negative effects of use of smartphone for work purposes in off-job hours on work-life conflict [38]. These factors demonstrate the key role of individual preferences in whether smartphone use during off-job hours leads to work-life conflict, and the impact it may (or may not) have.

This review resulted in recommendations for employers and line managers (Fig 3) which were developed with stakeholder input during review Stage 6.

## Limitations of included studies

The limitations of included studies mainly relate to the study design and the measurement of smartphone use for work-related purposes during off-job time. Most of the studies (19 of 23) relied on cross-sectional designs, and there was only one study that explored changes over time in a longitudinal design. This inhibits the establishment of causal relations among variables [97]. Of the remaining studies, four adopted a diary-entry design [23,45–47], one

**What is the problem?**

- When employees use smartphones for work purposes in non-work hours this can cause problems for some employees in balancing their work and home lives.

- For some employees, there can be negative impacts such as job stress and strain, and sleep disturbances.

- This could lead to reduced job satisfaction, burnout, and intentions to leave.

**What can employers do?**

- **Develop workplace policy** relating to work-life balance. To include 'availability' policy and where appropriate, discouragement of work-related smartphone use outside regular work hours (recognising needs of the job role, and individual preferences).

- **Encourage open conversations** with employees about their (and the organisation's) expectations of availability outside of working hours.

- **Enhance employee skills for 'psychological detachment' from work** by creating boundaries around the use of smartphones /email outside of working hours, where appropriate and mutually agreed.

**What can employers do?**

- **Provide training for line managers** about the potential negative effects on employees of smartphone use for work purposes during off-job hours. Consider the influence of leadership styles.

- **Raise awareness** of factors influencing decisions to use smartphones during non-work hours.

- These might include personal preferences, workplace norms, the nature of the job role, and the employees' level of control over their schedules and work.

- **Reduce** unnecessary job pressure and regularly **review** workloads.

- **Enhance** employees' job autonomy and **impart** full personal control over work schedules (where possible).

**Fig 3. Recommendations for employers and line managers (Photo 1 by Richard Rodrigues Photo 2 by Amy Hirschi, Photo 3 by Luis Villasmil; all on Unsplash).**

adopted a time-separated design [57], and one adopted a repeated measures design [38]. Although diary studies could be used for examining intra-individual changes across time, which is a component of longitudinal design [97], the included diary studies did not specifically provide evidence for intra-individual changes in participants, which deters the examination of causal relations. Also, the inclusion of only two measurement points in studies with a time-separated [57] or repeated-measures design [38] limit the determination of temporal relations among variables [98,99]. Importantly, our review demonstrates that papers focused on smartphone use operationalised the concept in different ways, with few studies measuring both functional (e.g., time spent answering emails) and perceptual (i.e., perceiving pressure to respond to email) aspects of smartphone use during off-job hours. Regarding the

measurement of smartphone use for work-related purposes during off-job time, almost all the studies used standardised measurement scales to assess work-life conflict (or the construct used to operationalise this). However, the use of self-report survey instruments increases vulnerability to recall bias. None of the included studies assessed the time spent on smartphones for work-related purposes during off-job time using objective data (e.g., recording screen time, such as the average minutes or hours using a smartphone). In addition, two of the included studies [42,45] used single-item self-constructed scales to assess the work-related use of smartphones thereby, inhibiting the determination of their internal consistency.

## Review strengths and limitations

Regarding study strengths, this scoping review involved stakeholder consultation which is an optional stage in the Arksey and O'Malley [31] framework. The review utilised pre-defined inclusion and exclusion criteria, a comprehensive and timely search strategy (searches up to date as of November 2023), pre-testing of all screening and data characterisation forms and quality appraisal. While quality appraisal is not an essential component of (or consistently included in) scoping reviews its inclusion addresses a known limitation of the scoping review method [100]. It was conducted and reported using a published methodological framework, quality assessment checklist and PRISMA-ScR reporting guidelines. At least two researchers were involved in each stage; there was independent and blind assessment of a random 20% of abstracts and full texts, with high inter-rater reliability. In terms of limitations, although we searched many databases which captured relevant papers in the social sciences (e.g., in the fields of psychology, business and management), the review may have missed some published studies through exclusion of databases in other disciplines (e.g., biomedical), grey literature, study protocols, and studies published in a language other than English. We intentionally excluded qualitative studies due to the nature of our research question and study aims, however, a qualitative or mixed-methods review may provide additional insights into this subject area.

## Review implications for research and practice

Studies in this review were conducted in nine countries although one third were conducted in the USA and there was only one study from the UK. There is scope for further research in other geographical regions, particularly those countries with the highest number of smartphone users (China: 974 million, India: 659 million [101]) and the highest smartphone penetration rates (France: 82.6%, UK: 82.2%, Germany, 81.9% [102]).

In this review, most studies found a significant association between increased use of smartphone for work purposes in off job-hours and increased work-life conflict. Findings from the review suggest that organisations should provide training for line managers about work-life conflict (or work-life balance) and the potential negative effects on employees of smartphone use for work purposes during off-job hours. Future research could focus on the co-creation of such line manager training with managers and other stakeholders (e.g., employer and employee representatives, professional bodies, trade unions). This training could be implemented and evaluated with managers from diverse employment settings and sectors, to explore outcomes for managers' knowledge and skills, and employees' perceptions of work-life conflict. Based on review findings, implications for practice were generated in collaboration between the study team and the interprofessional stakeholder group. While employers may wish to advocate for reduced use of smartphone for work purposes in off job-hours to reduce the risk for work-life conflict, should this be challenging due to the nature of the job role or individual preferences, then enhancing skills for psychological detachment may be one

approach to reducing or managing work-life conflict. The most appropriate mechanisms for achieving this could be explored in future evidence-reviews or qualitative research. Line managers should seek to reduce unnecessary job pressure and regularly review workloads to reduce unnecessary work-related smartphone use during off-job hours. Managers could review their leadership styles, aim to lead by example, and create a positive workplace culture in which they can have open conversations with employees about their (and the organisation's) expectations of availability outside of working hours, as well as their own and employees' segmentation preferences. More research is needed to explore the outcomes of open conversations in the workplace, and psychological safety climate, on individual and organisational outcomes.

Enhancing employees' job autonomy and imparting full personal control over work schedules may help to reduce negative impacts of smartphone use during off-job hours. This may help employees to manage or prevent work-life conflict where it is, or could be, experienced.

## Supporting information

**S1 Table. PRISMA checklist.**
(DOC)

**S2 Table. Operationalisation and measurement of smartphone technology and work-life conflict in included studies, and additional study variables.**
(DOCX)

**S3 Table. Summary of findings: The association between the use of smartphone technology for work in off-job hours and work-life conflict.**
(DOCX)

**S4 Table. Variables moderating or mediating the relationship between smartphone-use and work-life conflict.**
(DOCX)

**S5 Table. Summary of findings examining work-related wellbeing, attitudes, and work behaviours as outcomes.**
(DOCX)

**S1 Text. Example search strategy for PsychINFO.**
(DOCX)

## Acknowledgments

The authors thank Lana Delic for early input into the development of the research question and approach, and the interprofessional stakeholder group for contribution to Stages 1, 5 and 6.

## Author Contributions

**Conceptualization:** Holly Blake, Juliet Hassard, Jasmeet Singh.

**Data curation:** Jasmeet Singh.

**Formal analysis:** Juliet Hassard, Jasmeet Singh, Kevin Teoh.

**Investigation:** Holly Blake, Juliet Hassard, Jasmeet Singh, Kevin Teoh.

**Methodology:** Holly Blake, Juliet Hassard, Jasmeet Singh.

**Project administration:** Jasmeet Singh.

**Writing – original draft:** Holly Blake, Jasmeet Singh.

**Writing – review & editing:** Juliet Hassard, Kevin Teoh.

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
