## [Decision Letter · Decision Letter 0]

17 May 2024

PDIG-D-24-00079

Work-related smartphone use during off-job hours and work-life conflict: a scoping review.

PLOS Digital Health

Dear Dr. Blake,

Thank you for submitting your manuscript to PLOS Digital Health. After careful consideration, we feel that it has merit but does not fully meet PLOS Digital Health's publication criteria as it currently stands. Therefore, we invite you to submit a revised version of the manuscript that addresses the points raised during the review process.

Please submit your revised manuscript within 60 days Jul 16 2024 11:59PM. If you will need more time than this to complete your revisions, please reply to this message or contact the journal office at digitalhealth@plos.org. Please include the following items when submitting your revised manuscript:

We look forward to receiving your revised manuscript.

Kind regards,

Baki Kocaballi

Section Editor

PLOS Digital Health

Journal Requirements:

1. Please provide separate figure files in .tif or .eps format only and remove any figures embedded in your manuscript file. Please also ensure that all files are under our size limit of 10MB.

Additional Editor Comments (if provided):

The paper titled "Work-related Smartphone Use During Off-Job Hours and Work-Life Conflict: A Scoping Review" is well-written and interesting. In order to enhance the rigor and comprehensiveness of their study, it is imperative for the authors to meticulously detail their methodology, offering transparency and clarity to their research process. Additionally, the authors can explore deeper certain concepts. Moreover, it is essential for the authors to frame their research in a context that highlights its potential impact on policymaking and future investigations - by elucidating the practical implications and broader significance of their findings, the authors can underscore the relevance and urgency of their work in informing policy decisions and guiding future research. Please address reviewers' comments.

Reviewers' comments:

Reviewer's Responses to Questions

**Comments to the Author**

1. Does this manuscript meet PLOS Digital Health’s publication criteria? Is the manuscript technically sound, and do the data support the conclusions? The manuscript must describe methodologically and ethically rigorous research with conclusions that are appropriately drawn based on the data presented.

Reviewer #1: Partly

Reviewer #2: Yes

Reviewer #3: Yes

2. Has the statistical analysis been performed appropriately and rigorously?

Reviewer #1: N/A

Reviewer #2: N/A

Reviewer #3: Yes

3. Have the authors made all data underlying the findings in their manuscript fully available (please refer to the Data Availability Statement at the start of the manuscript PDF file)?

Reviewer #1: No

Reviewer #2: Yes

Reviewer #3: Yes

4. Is the manuscript presented in an intelligible fashion and written in standard English?

Reviewer #1: Yes

Reviewer #2: Yes

Reviewer #3: Yes

5. Review Comments to the Author

Reviewer #1: This is work that was carried out using the Prisma methodology. In general, the work meets the requirements of this methodology. A better systematization of Table 1 Characteristics of Included Studies is suggested

line 246 (n = 1 (37); one-week data collection period),

Reviewer #2: This is an interesting scoping review on the use of the smart phone in off-work hours. The topic is important and timely and the paper is very well written. I have a few comments:

- Although it is described like this in the framework of Arksey and O’Malley, I assume that Step 1 would be redundant as the research question has been clear before the authors selected the method. Also, aims are described in the introduction of the paper which differ from the research question in Step 1. 

- Could you please describe how you selected the databases? Why did you not include Pubmed for example?

- Why did you exclude qualitative studies?

- Did you include protocols and ongoing research projects or only completed studies?

- Please include some more details about the stakeholders. How were they selected? What impact did the stakeholder consultations have on the results?

- Table 1 is quite extensive. Could some of the criteria be broken down and reported in a more structured way?

Reviewer #3: The study follows a clear structure and methodological framework, and the findings are well-described. 

Here are a few suggestions for improvement:

i) Some terms, such as "psychological detachment from work," could be defined or explained briefly for readers who may not be familiar with them.

ii) A section on future research directions based on the findings of this scoping review can be added.

iii) It's good that Google Scholar was also used for additional articles, but a brief mention of the criteria used for selecting articles from Google Scholar to maintain consistency with the articles obtained from other databases.

6. PLOS authors have the option to publish the peer review history of their article (what does this mean?). If published, this will include your full peer review and any attached files.

**Do you want your identity to be public for this peer review?** For information about this choice, including consent withdrawal, please see our Privacy Policy.

Reviewer #1: No

Reviewer #2: Yes: Tanja Stamm

Reviewer #3: No

---

## [Editor Report · Decision Letter 1]

15 Jun 2024

Work-related smartphone use during off-job hours and work-life conflict: a scoping review.

PDIG-D-24-00079R1

Dear Professor Blake,

We are pleased to inform you that your manuscript 'Work-related smartphone use during off-job hours and work-life conflict: a scoping review.' has been provisionally accepted for publication in PLOS Digital Health.

Best regards,

Raquel Simões de Almeida, PhD

Academic Editor

PLOS Digital Health

I'm pleased to inform you that you have successfully addressed all the reviewers' comments and significantly improved the quality of the manuscript. In my perspective, the paper should be accepted pending minor changes. Please see them below:

Line 88 - the full stop must appear after the reference “sensitive to existing inequalities. (15)”

Line 227 – “and” before stage 5 can be removed “in the literature and/or practice) and Stage 5 (interpreting findings) and Stage 6”

Table 1 - the acronym NR must be specified in the caption

Fig. 3 - square symbols are appearing instead of dots at the beginning of each idea

Supplementary File 3 – the acronym N/A must be specified in the caption